# Collapse and rescue of cooperation in evolving dynamic networks

Erol Akçay [1]

The evolutionary dynamics of social traits depend crucially on the social structure of a population. The effects of social structure on social behaviors are well-studied, but relatively little is known about how social structure itself coevolves with social traits. Here, I study such coevolution with a simple yet realistic model of within-group social structure where social connections are either inherited from a parent or made randomly. I show that cooperation evolves when individuals make few random connections, but the presence of cooperation selects for increased rates of random connections, which leads to its collapse. Inherent costs of social connections can prevent this negative feedback, but these costs can negate some or all of the aggregate benefits of cooperation. Exogenously maintained social inheritance can mitigate the latter problem and allow cooperation to increase the average fitness of a population. These results illustrate how coevolutionary dynamics can constrain the long-term persistence of cooperation.

---

[1] Department of Biology, University of Pennsylvania, Philadelphia, PA 19104, USA. Correspondence and requests for materials should be addressed to E.A. (email: eakcay@sas.upenn.edu)

Cooperation is easy to evolve. In the last half century, we have discovered that there are myriad ways natural selection can favor organisms providing benefits to each other. These pathways include population structure[1], phenotypic feedbacks[2,3], payoff synergies[4], partner choice[5], among others e.g., see reviews in refs. [6–8] When operating together, these pathways to cooperation can reinforce[6,9–11] or counteract each other[12]. This extensive literature overwhelmingly tries to explain how cooperation can persist in the face of conflicts of interests. But with so many ways cooperation might be selected for, it is worth asking why cooperation is not even more prevalent.

The answer to this question lies in how the conditions leading to cooperation themselves evolve, i.e., how selection acts on the setting in which the interaction takes place i.e., the payoff structure, interaction network, etc.[13,14], and how the setting coevolves with cooperation. This question has recently been garnering attention. An emerging common thread is that these coevolutionary processes might impose inherent limits to the maintenance of cooperation in the long term. For example, in a model of evolution of incentives for cooperation, Aky and Roughgarden[14] showed that incentives that favor cooperation may invade but not fix, leading to stable polymorphisms where cooperation and defection are both maintained in the population. In another model of payoff evolution, Stewart and Plotkin[15] showed a different kind of dynamic self-limitation: when cooperation is established in the population, it tends to drive the evolution of payoffs for cooperation so high that the temptation for defecting becomes overwhelming, leading to the collapse of cooperation. In a model incorporating environmental feedbacks that affect the payoffs, Weitz et al.[16] showed that negative feedbacks between social strategies and environmental variables that favor them can create oscillations between cooperative and non-cooperative outcomes. More recently, Mullon et al.[17] showed that in settings where dispersal and cooperation coevolve, selection might result in stable polymorphisms where non-cooperators persist by evolving higher dispersal rates.

One major factor in the evolution of cooperation is the social structure of groups (i.e., who interacts with whom), represented by social networks[18,19]. Social networks and variation in individuals' positions in them are shown to affect important life history traits such as reproductive success[20], survival[21,22], infant survival[23], as well as selection on particular behaviors[24], and resilience of social groups[25]. Yet despite the emerging evidence about the importance of dynamic fine-scale social structure, it has not yet been integrated fully into social evolutionary theory, where most models deal with special kinds of networks e.g., lattice structured[26], fixed networks[27–29], dynamic networks with random connections[30,31], or shifting connections amongst a fixed set of

individuals[32–34]. A recent study by Cavaliere et al.[35] comes closest to the current work: they model the evolution of cooperation on a dynamic heterogenous network structured by pure social inheritance (as defined below), though without a feedbacks between the evolutionary dynamics of social structure and cooperation (see Discussion for more). Although each of these modeling approaches captures important aspects of how population structure affects cooperation, we know relatively little about how social traits might evolve in more realistic dynamic social networks, and how these traits might feed back on the structure of networks.

This gap is in part caused by the lack of a generally applicable model for network dynamics that can capture important features of social networks and variation therein. Recently, Ilany and Akçay[36] proposed such a model, where social ties are formed by a mixture of individuals "inheriting" connections from their parents, i.e., connecting to their parents' connections, and randomly connecting to others. They showed that this simple process of social inheritance can capture important features of animal networks such as their degree and clustering distributions as well as modularity. Importantly, the animal networks investigated by Ilany and Akçay tended to have relatively high probabilities of social inheritance, while having low (but non-zero) probability of random linking. These findings suggest that the social inheritance process is a good candidate for modeling the fine-scale dynamics of animal social networks and the evolution of social behaviors on them. Importantly, they raise the question of how social inheritance affects the evolutionary dynamics of social behaviors, and how social inheritance coevolves with these behaviors.

In this paper, I present a computational model of the evolution of a cooperative behavior on a dynamic network that is assembled through social inheritance. I find that cooperation evolves when the probability of random linking is low, mostly independently of the probability of social inheritance. However, when these two linking probabilities themselves coevolve with cooperation, I show that in cooperative populations, probabilities of random linking are selected to increase, which in turn leads to the collapse of cooperation. This result highlights a new way in which some forms of cooperation can inherently be self-limiting. I then show that costs of making and maintaining social links can counteract the self-limiting feedback through the evolution of social structure. At the same time, costly links lead to unexpected non-monotonic patterns in long-term frequency of cooperation. Overall, my results shed light on new kinds of evolutionary feedbacks between traits that structure social networks and the social behaviors that evolve on them.

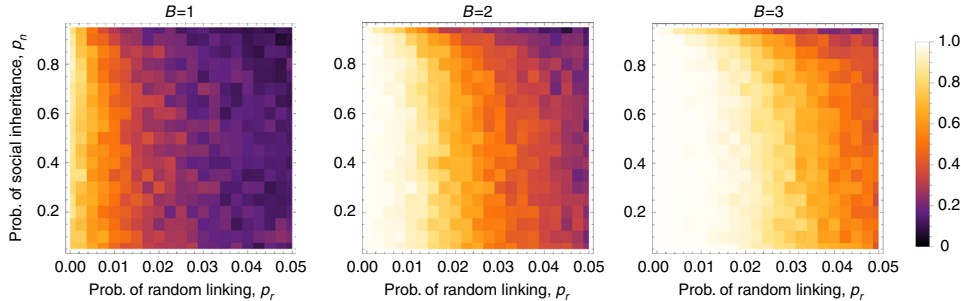

**Fig. 1** The frequency of cooperation at different values of the benefit $B$ as a function of the probabilities of social inheritance, $p_n$, and random linking, $p_r$, when these linking probabilities are kept fixed in the population. For each combination of linking probabilities, the simulation was run for 500 generations (each generation equals $N$ death-birth events). I recorded the frequency of cooperation at intervals corresponding to $N$. The color in each cell depicts the average frequency of cooperation over the last 400 generations for 100 replicate simulations. Parameters are $N = 100$, $C = 0.5$, $D = 0$, $\mu = 0.001$, $\delta = 0.1$, $C_{link} = 0$

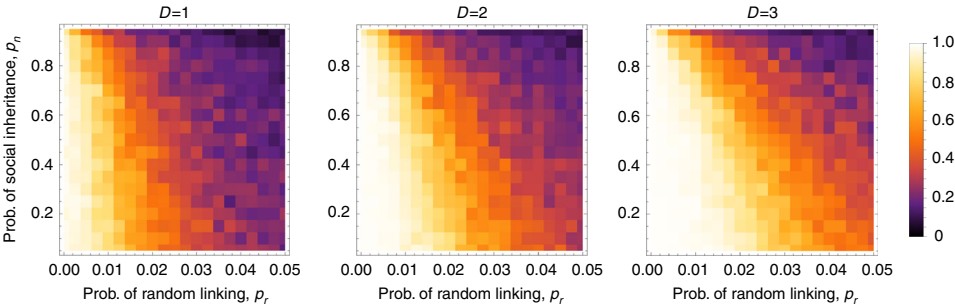

**Fig. 2** The effect of synergism on the frequency of cooperation as a function of the probabilities of social inheritance, $p_n$, and random linking, $p_r$. The simulations are run as in Fig. 1. Parameters are $N = 100$, $B = 1$, $C = 0.5$, $\mu = 0.001$, $\delta = 0.1$, $C_{link} = 0$

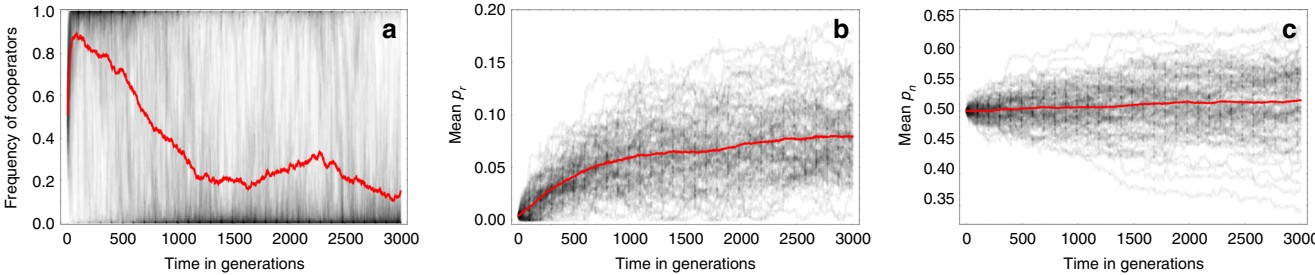

**Fig. 3** Density of trajectories of frequency of cooperation (**a**), mean $p_r$ (**b**), and mean $p_n$ (**c**) from 200 replicate simulations, depicting coevolution of the linking probabilities with cooperation. In each panel, darker regions correspond to a higher proportion of replicate trajectories passing through that point. The red curve in each panel depicts the mean of 100 trajectories. Each trajectory starts with $p_r = 0.001$ and $p_n = 0.5$, which for these parameter values favors cooperation. Accordingly, cooperation is established quickly after starting at frequency 0.5, as indicated by the dark spot at the upper left corner of **a**, which indicates most trajectories initially converge to high frequency of cooperation. However, this is followed by an increase in the mean $p_r$ value of the population (**b**), and cooperation soon collapses, with trajectories increasingly spending time near zero frequency of cooperation. After cooperation has collapsed, $p_r$ continues its upward trajectory but under somewhat relaxed selection. In contrast to $p_r$, there is no strong directional selection on $p_n$, and trajectories spread out in both directions from the initial value in **c**. Parameter values are $N = 100$, $B = 2$, $C = 0.5$, $D = 0$, $\mu = 0.001$, $\mu_l = 0.01$, $\delta = 0.1$, $\sigma_n = 0.01$, $\sigma_r = 0.01$

## Results

**Overview of model**. My model builds on Ilany and Akçay's[36] by adding selection caused by social interactions on a dynamic, binary, and undirected network. I assume a death–birth process, where at each time step, a random individual is selected to die, and another individual is selected to reproduce to replace them. The newborn individual makes social connections as follows: (i) it connects to its parent with certainty, (ii) it connects to each individual that is connected to its parent (at the time of birth) with probability $p_n$, and (iii) it connects to each individual that is not connected to its parent (at the time of birth) with probability $p_r$. Ilany and Akçay[36] showed that this basic model (with no selection) can capture important aspects of social structure in the wild. Here, I add selection caused by social interactions, where the probability of being selected to reproduce is an increasing function of the payoffs individuals obtain from their social interactions. In particular, I assume there are two types of individuals, cooperators and defectors: cooperators provide their connections a total benefit $B$, divided equally amongst all their connections, and pay a cost $C$; defectors pay no costs and provide no benefits. I also consider scenarios where two connected cooperators each get a synergistic benefit $D$, divided by the product of their degrees. Details of the model are given in the Methods section. Below, I first consider $p_n$ and $p_r$ to be fixed and the same for every individual. Then, I let the linking probabilities $p_n$ and $p_r$ be heritable and vary between individuals, so that they evolve according to their fitness consequences.

**Fixed linking probabilities**. I first consider the fate of a cooperation allele in groups that have fixed probabilities of random

linking $p_r$ and social inheritance $p_n$, and no costs to linking. I find that cooperation is maintained only under relatively low $p_r$ (Fig. 1). Interestingly, for most of its range $p_n$ makes relatively little difference in the long-term frequency of cooperation. This indifference breaks down at very high levels of $p_n$, which disfavors cooperation. These results represent relatively straightforward cases of kin selection: low random linking and not too high social inheritance produces high assortment between connected individuals, so that cooperators benefit more from interactions with other cooperators (Supplementary Fig. 1). Higher probability of random linking reduces the assortment, as might be expected. Intuitively, we might expect higher social inheritance to increase assortment and therefore favor cooperation, but in fact, the effect of social inheritance on assortment is neutral or negative (at high $p_n$). This is because in the current model offspring have the same probability of inheriting all connections their parents make, regardless of whether those connections were inherited or made randomly. At high social inheritance, however, networks evolve to be very densely connected which reduces the potential for assortment[36] (Supplementary Fig. 1). In more densely connected networks, the average benefit per link obtained from a cooperative partner also decreases due to the dilution effect (Eq. (1)). This also works against cooperation (compare with results for the prisoner's dilemma game with constant benefit per link in Supplementary Note 2).

With positive synergism between cooperators the picture changes slightly. As expected, stronger synergistic interactions (higher $D$) make cooperation possible for a larger range of $p_n$ and $p_r$ value (Fig. 2), as synergism generates benefits that are only available to other cooperators[4,37]. However, this added benefit is

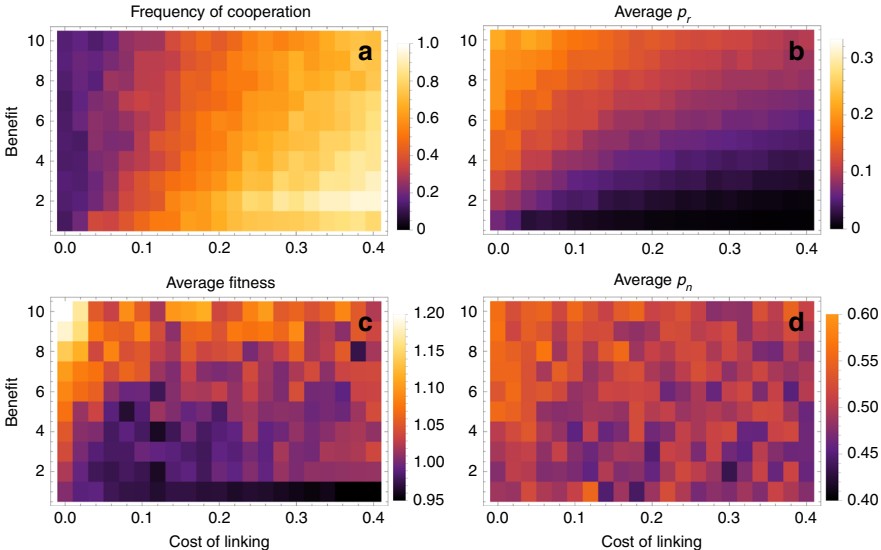

**Fig. 4** Mean frequency of cooperation (**a**), $p_r$ (**b**), $p_n$ (**d**), and average fitness (**c**) over ten replicate simulations, averaged across time, as a function of the benefit from cooperation and the cost of linking. Each simulation was initiated with $p_r = 0.0001$, $p_n = 0.5$, frequency of cooperation at 0.5, and run for $10^5$ generations ($10^7$ time steps). Averages over the final $8 \times 10^4$ generations (sampled once each generation) are shown. Note that these simulations are run over longer time-periods than the ones in Fig. 1 to give the system time to explore the larger (and continuous) state space that includes the evolving linking traits. Parameter values are $N = 100$, $C = 0.5$, $D = 0$, $\mu = \mu_l = 0.001$, $\sigma_n = \sigma_r = 0.01$, $\delta = 0.1$

mainly realized at low levels of social inheritance, when the average degree of individuals is low and therefore the synergistic benefits are less diluted. As a result, with positive synergism, increasing both social inheritance and random linking probabilities (both of which increase the average degree of individuals) favors defection.

**Coevolution of linking probabilities undermines cooperation**. Next, I let the linking probabilities $p_n$ and $p_r$ coevolve with cooperation. Figure 3 shows a collection of sample trajectories that start with a low probability of random linking. Cooperation is quickly established in the population, but once it is established, it creates selection for the probability of random linking, $p_r$, to increase. Increased $p_r$ in turn reverses selection on cooperation, and defection is established again in the population. These dynamics reveal that cooperation is self-limiting when the social structure co-evolves with it: once cooperation establishes in a population, it creates selection against the social structure that allowed it to evolve in the first place. The intuition behind this result is quite simple and general: in a cooperative population, it pays to make connections with any individual, since there is likely to be a benefit to be had from that connection. Therefore, individuals with higher probability of random linking (and thus, more connections) fare better in a cooperative population. This leads to a population with high probability of random linking, where we know cooperation cannot persist.

**Costs of linking can rescue cooperation**. One possible mechanism that can counteract these dynamics is when making and maintaining social links is inherently costly, regardless of one's phenotype or that of partners. Such costs can counteract the incentive to seek out more connections, and prevent the linking probabilities (specifically, $p_r$) from crossing the threshold beyond which cooperation cannot be sustained. Figure 4 confirms that costs of social connections can prevent cooperation from limiting itself: for a given value of benefit $B$, as the cost of linking, $C_{\text{link}}$ increases, the long-term average frequency of cooperation tends to increase. Interestingly however, this long-term average displays a non-monotonic pattern in $B$ for moderate to high $C_{\text{link}}$: as

$B$ increases from low values, cooperation at first becomes more prevalent, as one might intuitively expect. In contrast, at higher values of $B$, making cooperation more beneficial reduces its long-term frequency. This "paradox of enrichment" (no relation to the one observed in prey–predator dynamics[38]) is another manifestation of the self-limiting nature of cooperation in dynamic networks: as the benefit from cooperation increases, so does the incentive to make random links in a cooperative population. Therefore, $p_r$ evolves to higher values, which eventually undermines cooperation. Equivalently, a higher cost of linking is required to keep $p_r$ low and maintain cooperation. This effect can be seen by looking at the average $p_r$ (Fig. 4b), which increases with $B$ for a given cost of linking. We observe this non-monotonicity of cooperation and increase of $p_r$ with the benefit $B$ in both strong and weak selection (compare with Supplementary Fig. 6).

A final paradoxical result in Fig. 4c is that even when cooperation is sustained, the costs incurred may be too high, such that the average fitness of the population (calculated using Eq. (14)) can actually be lower in a more cooperative population than a less cooperative one (e.g., compare $B = 2$ and $C_{\text{link}} = 0$ and $C_{\text{link}} = 0.4$ in Fig. 4c). More generally, for a given level of benefit $B$, the mean fitness of a population follows a non-monotonic pattern with the costs of linking: the mean fitness first decreases and then increases with $C_{\text{link}}$. In other words, even though cooperation can be rescued by costs of social connections, the victory may prove pyrrhic.

When the main benefits from cooperation come from synergistic payoffs, cooperation tends to be stable when links are also costly, as shown in Fig. 5 for weak selection (strong selection yields similar results; see Supplementary Note 1). Here, synergism and costs of linking interact positively: for a given (non-zero) cost of linking, increasing synergy increases the frequency of cooperation, and vice versa. As a result, the more synergistic the payoffs, the lower the cost of linking required to maintain cooperation. Like the no-synergism case, cooperation tends to be accompanied by low $p_r$ and $p_n$. Furthermore, $p_n$ and $p_r$ display monotonically decreasing patterns in both synergism and cost of linking 5(b, d), resulting in very sparsely connected

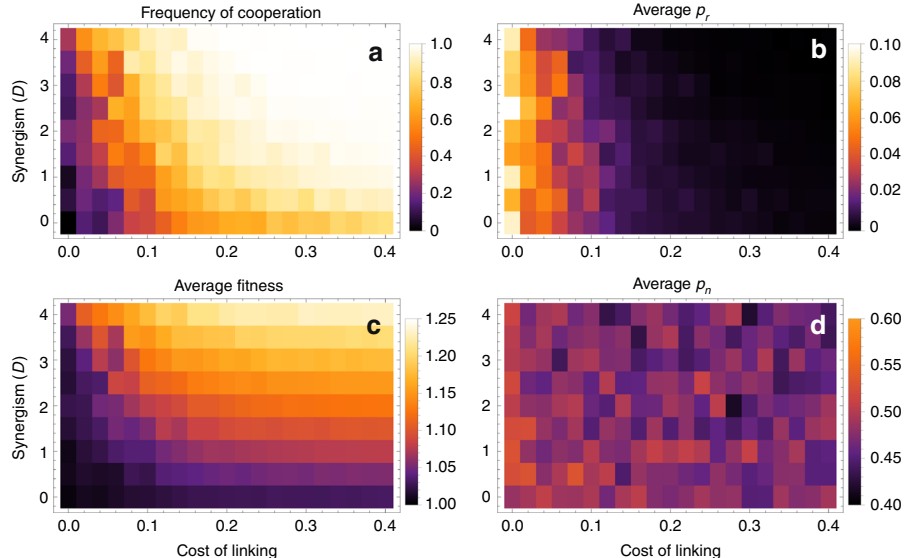

**Fig. 5** Effect of synergistic payoffs on cooperation (**a**) and the probabilities of random linking $p_r$ (**b**), and social inheritance $p_n$ (**d**), and the mean fitness (**c**). Simulations and averages were performed as in Fig. 4. Here, $B = 1$; the other parameters as in Fig. 4

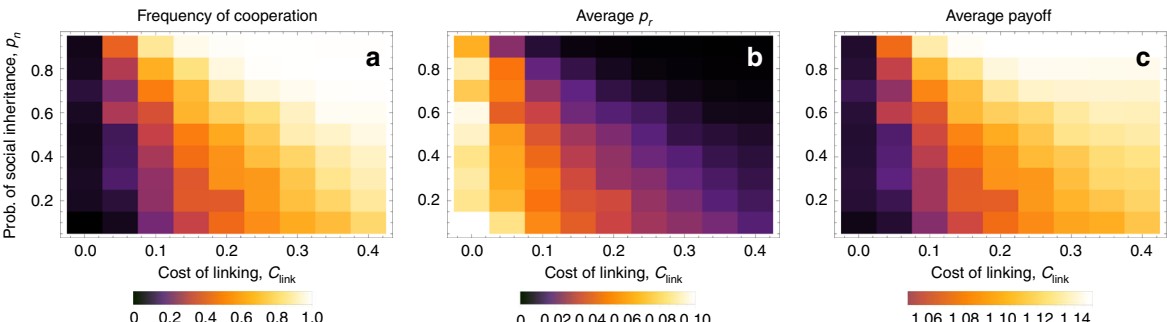

**Fig. 6** Effects of exogenously fixed probability of social inheritance $p_n$ when the probability of random linking $p_r$ is evolving. **a** Depicts the long-term average frequency of cooperation, **b** the average $p_r$, and **c** average payoff. Here, $B = 2$, $C = 0.5$, $D = 0$, $\mu = \mu_{link} = 0.001$, $\sigma_r = 0.01$

networks (Supplementary Fig. 5). This is because synergistic benefits are inversely proportional to the product of degrees, and therefore selection on reducing the mean degree of cooperators is strong. In general, synergistic payoffs, together with some costs of linking promote cooperation and increase mean fitness 5(c), but result in very sparsely connected networks.

**Exogenously high social inheritance can rescue cooperation.** One surprising aspect of the preceding results is that there is relatively little selection on the probability of social inheritance $p_n$ due to feedbacks from the social trait, in contrast to the probability of random linking, $p_r$. This suggest that social inheritance might predominantly evolve due to other selection pressures or as a pleiotropic consequence of group living (e.g., due to offspring passively being in proximity of their parent's connections). This raises the question of how cooperation and $p_r$ fare when social inheritance is fixed by an exogenous factor. Figure 6 shows that when $p_n$ is fixed exogenously but $p_r$ is left free to co-evolve with cooperation, high social inheritance can maintain high frequency of cooperation in conjunction with costly linking. This result may appear to be in contradiction with my first results above when both linking traits are fixed, where higher $p_n$, holding $p_r$ fixed, reduces cooperation. The contradiction is resolved by observing that when $p_n$ is high, and links are costly, $p_r$ will evolve to be lower, because the higher $p_n$ the higher marginal cost of higher $p_r$ due to the steeper increase in the expected mean degree of an

individual Figure 2 of ref. [36]. This means that for a given benefit from cooperation, the marginal cost of increased random connections becomes equal to the marginal benefits at a lower level of $p_r$, which favors cooperation. Biologically, this means that high social inheritance, if evolved (and maintained) for reasons other than cooperative benefits, can also function to sustain cooperative populations (see Discussion for more).

## Discussion
I use a simple dynamical network model that is able to reproduce important characteristics of animal social structure based on social inheritance[36], and investigate how a social behavior such as cooperative investments can evolve under such dynamics. My results show that cooperation tends to evolve under low rates of random linking. Interestingly, the probability of social inheritance makes little difference for most of its range, except at high levels, where it disfavors cooperation, which is contrary to the intuition that inheriting more links should make networks more assortative. This intuition, however, comes from an implicit assumption that the overall connectivity of a network stays the same, so inheriting more links means making fewer random connections. This is not the case in the current model. More interesting patterns arise when linking probabilities co-evolve with cooperation. Cooperation readily evolves when networks start with low levels of random linking, but once it does, selection increases the rate of random linking, undermining cooperation. Costs of linking can

counteract this self-limitation of cooperation, however, they also have to contend with a second kind of self-limitation, where as cooperation becomes more beneficial, the level of linking costs required to maintain cooperation at high frequency increases and may negate some or all of the benefit from cooperation. Exogenously maintained high levels of social inheritance or synergistic payoffs from cooperation can help overcome this self-limitation problem, allowing cooperation to establish with relatively low costs of linking. But in the case of synergistic payoffs, the resulting networks are sparsely connected with low $p_n$ and $p_r$.

These results add to a growing theoretical literature that is uncovering cases where evolutionary dynamics lead to the establishing of cooperation, only to undermine it through altering the conditions that select for it in the first place. Previous results uncovered such negative feedbacks operating through the payoff structure of a game, whether by direct evolution of payoffs[14,15] or through environmental feedbacks that alter the returns from different strategies[16]. Here, I identify a different kind of dynamical feedback between cooperation and the setting in which it evolves. By focusing on the interplay between a simple yet realistic model of network dynamics and social behaviors, I show that the structure of the society that favors cooperation can itself fall victim to cooperation. The logic behind this phenomenon applies generally beyond the current model: cooperation tends to be favored when population assortment is high. But regardless of the details of the process of acquiring connections, in cooperative populations, connections with most individuals are beneficial, and therefore individuals in such populations would be selected for making more connections indiscriminately. All else being equal, this would lead to more mixing in the population, which in turn disfavors cooperation. Thus, coevolution of the network structure with social traits such as cooperation sets up a fundamental negative feedback that has not previously been recognized. Furthermore, the negative feedback is stronger with higher benefits of cooperation, which increases the temptation to link randomly at high rates.

Costs to making and maintaining connections might counteract this negative feedback by reducing selection for increased probability of making random connections. Accordingly, I find that with high enough costs, cooperation can be maintained. Maintenance of social bonds in many animal and human societies involves costly investments[39,40], which in many cases are not beneficial to either party. Previous theory hypothesized that such costs might serve signal partner quality[41] or building trust[42]. My results show that regardless of their function at the level of the pair of individuals, costs of maintaining bonds shape the social structure of a group in a way that facilitates cooperation. Therefore, it is possible that such costs can evolve through cultural or genetic selection at the group level as a group-level adaptation that sustains selection for cooperation within groups. However, this expectation is tempered by the fact that even when cooperation is rescued, the costs of linking may be too high for cooperation to be a net benefit to the population on average (Fig. 4c).

A potential way for populations to avoid such a pyrrhic victory for cooperation is when social inheritance is kept high by factors exogenous to the current model. Then, costs of linking can promote cooperation and high payoffs (Fig. 6). This happens because high social inheritance effectively increases the costs of random linking, as these links are more likely to get inherited, and therefore increase the average degree of a lineage[36]. Thus, for a given cost of linking, higher $p_n$ means higher marginal costs of increasing $p_r$, which in turn means that $p_r$ evolves to a lower level, which maintains more cooperation. I furthermore show that at least under weak selection, coevolving cooperation will not strongly oppose selection on $p_n$ due to other factors. Taken

together, these results suggest a potential pathway to cooperation where high social inheritance can evolve for reasons other than cooperation, including possibly group-level selection on behavioral (or institutional) traits that favor both social inheritance and some costs of linking, thereby favoring within-group cooperation. Exploring the multi-level evolutionary dynamics of linking traits and their costs is likely to yield further interesting insights.

Finally, it is worth noting that the connection costs do not rescue cooperation in the Prisoner's Dilemma game (see "Evolving linking probabilities" in Supplementary Note 2), where the benefits and costs from cooperation increase linearly with degree. This happens because connection links in such a scenario only serve to effectively increase the costs of cooperation. If cooperation evolves for a given level of linking costs, it implies that the benefit of being connected to a cooperator exceeds the joint costs of cooperation and linking. Therefore, random linking will necessarily increase and cooperation will collapse. This represents another way in which cooperation can be too beneficial for its maintenance.

A potential way to avoid these fundamentally self-limiting dynamics of cooperation is partner choice[5], i.e., preferentially interacting with cooperators or avoiding defectors. Papers by Pacheco et al.[30] and Santos et al.[31] provide models of evolution of cooperation through partner choice in dynamic networks. In these models, players make and break connections with each other at rates that depend on the type of the partners. These models show that cooperation can evolve and be stable in dynamically changing networks. However, these models consider type-dependent linking rates as exogenously fixed, and do not consider how they might co-evolve with cooperation. When the coevolutionary dynamics are considered, it is likely that we would recover the self-limiting nature of cooperation in these models as well. This is because at highly cooperative populations, there would be little need to maintain differential connection rates, which means that they would erode, setting up the stage for the collapse of cooperation. Previous models have shown that in pairwise interactions adequate mutation rate[43] or immigration from a source population with high variation[44] is required to maintain choosiness and thus cooperation. How the dynamics of partner choice operate in a network context remains to be explored.

Another mechanism that can maintain cooperation is direct reciprocity between interacting individuals[2,3]. While I do not model reciprocity explicitly, we know that in pairwise interactions, the effects of reciprocity can be accounted for by a synergistic payoff function, where reciprocators achieve an extra benefit not available to non-reciprocators[10,37]. I find that synergistic payoffs such as those that might be expected from reciprocal cooperation tend to (unsurprisingly) favor cooperation, but they are still subject to the self-limitation problem. However, with reciprocity, the self-limitation problem is resolved more easily and costs of linking that prevent high random linking act in concert with synergistic benefits, rather than antagonistically like they do with additive benefits. This pattern is consistent with previous results that show behavioral responses and population structure tend to act in synergy with each other[9–11]. One caveat here is that strong synergism tends to select for sparsely connected networks with low $p_n$ and $p_r$, due to the dilution effect on synergistic benefits.

It is interesting to relate the results in network-structured populations presented here to the rich literature on spatially structured models of social evolution, where dispersal determines a newborn's social partners. For example, Koella[45] modeled a scenario where investment into cooperation can coevolve with dispersal and interaction distances on a lattice. When dispersal

and interaction were fixed and local, cooperation readily evolved, but when dispersal distance coevolves with cooperation, defectors evolve long-distance dispersal which leads to reduced investment by cooperators, analogous to what happens on my model with increased random linking and subsequent collapse of cooperation. When interaction neighborhood also evolves, cooperators and defectors can coexist in spatially alternating bands, with cooperators interacting hyper-locally. Another paper, by Smaldino and Schank[46], explores how movement traits coevolve with cooperation in a spatially structured population where individuals move around a lattice looking for partners. They find that cooperation is more successful with less movement, which in terms of the interaction networks would correspond to most individuals staying near their parents and therefore making fewer random links and socially inheriting more. In contrast, when defectors moved a lot, they did better as they were able to find and exploit cooperator clusters. When cooperators and defectors used their respective best movement strategies, Smaldino and Schank observed persistently polymorphic populations. Likewise, Mullon et al.[17] found that in patch-structured populations, cooperators and defectors can coexist with the former evolving low dispersal rates and the latter high. These results are analogous to those I find in larger networks (see "Evolution in large networks" in Supplementary Note 1), where cooperators and defectors can coexists for periods of time with cooperators evolving low linking probabilities and defectors high. Another interesting result from a lattice-structured model to compare to the current paper is that of Smaldino et al.[47], who found that increasing the costs of cooperation can favor cooperation in the long-run. This result is some ways analogous to my finding that higher benefits can reduce the long-term frequency of cooperation, but happens due to a different mechanism: in Smaldino et al., high costs first cause most cooperators to die out, followed by defectors. Cooperators can then re-invade an empty landscape in clusters, which are subsequently surrounded by defectors but can get big enough to avoid the costs of being taken advantage of.

The closest existing model to the present one is one by Cavaliere et al.[35], who consider the evolution of cooperation in a population with fixed (and moderately high) social inheritance, but no random linking. Consistent with my results with fixed linking probabilities, their populations evolve to be mostly cooperative. However, Cavaliere et al. find in their simulations that cooperative populations were densely connected while defector networks are sparse. This pattern arises because in cooperative societies more connected individuals are selected for, especially since Cavaliere et al. assume that the benefit per link from a cooperator is constant rather than being diluted as 1/degree as in this model. Although Cavaliere et al. assume no variation in individuals' linking traits, individuals can still pass down their higher degree to their offspring through social inheritance, which happens in my model as well. However, since this effect is purely due to social inheritance, and not the evolution of linking traits themselves, it does not limit cooperation in the long-term. In contrast, once the average linking probabilities (especially $p_r$) evolve to high values in my model, the population spends more time in low-cooperation states. It is also worth noting that the change in network structure found by Cavaliere et al. requires relatively strong selection (e.g., very high absolute values of $B$ and $C$); under weak selection, the presence or absence of cooperation by itself has a relatively small effect on network structure. For the same reason, under weak selection and fixed linking probabilities, costly linking would also have little effect on network structure or the presence of cooperation, although with strong selection they can depress the connectivity of the network and therefore maintain cooperation for a wider range of (fixed) $p_n$ and $p_r$ values.

It is interesting to ask how the linking probabilities that favor cooperation compare to observed social networks in the wild. Ilany and Akçay[36] find that animal social networks tend to be characterized by moderate to high social inheritance, $p_n$ (0.5–0.8), and low $p_r$ (0–0.1). These linking probabilities are generally consistent with the presence of cooperation for a range of payoff parameters in my model. Therefore, my results suggest that conditions for cooperation might be met in the wild. When the linking probabilities themselves evolve, I find that the random linking probability, $p_r$, responds to different selective forces in an intuitive way: evolved $p_r$ decreases with increasing costs of linking, and increases with increasing benefits from cooperation. On the other hand, the social inheritance probability, $p_n$ seems to be somewhat less intuitive: at least under weak selection, $p_n$ behaves largely neutrally, exhibiting little sensitivity to benefits of cooperation or costs of linking (see "Results under strong selection" in Supplementary Note 1). Therefore, my results suggest that other factors that are not modeled here, such as obtaining support in social conflicts or between-group selection (as discussed above), might select for higher social inheritance. Alternatively, $p_n$ might be high simply as a by-product of the parent–offspring associations (offspring spending a lot of time with their parents and therefore the parents' connections).

In conclusion, my results show that the evolution of social traits such as cooperation can have unexpected consequences for the social structure that determines the direction of social selection. I identify a fundamental negative feedback that causes cooperation to be self-limiting through its effects on the social network structure. These results highlight the need to understand dynamic feedbacks between selection acting social traits and the environment in which they evolve. These feedbacks might help explain why not every cooperation problem in nature will be solved despite the myriad theoretical mechanisms available in principle, or why the solution might not always prove to be beneficial on the net. Focusing on these feedbacks will allow us to move beyond explaining how selection can favor cooperation in principle to predicting when the conditions that favor it are likely to exist.

## Methods

**Social interaction and fitness**. My model extends the previously published one by Ilany and Akçay[36] by adding selection caused by social interactions. To add selection, I assume that each individual can be of one of two types: cooperators and defectors. Cooperators provide a benefit $B$ to their partners (those that are connected to them on the network), distributed equally amongst all partners. In other words, if a given cooperator individual has $d$ connections, each of its partners acquire a benefit $B/d$ from it. Cooperators also pay a fixed cost $C$, regardless of the number of type of their connections. This game is a special case of the "coauthor game" of Jackson and Wolinsky[48]. Intuitively, it represents an interaction where cooperators have a fixed time or energy budget to help others e.g., by spending time grooming[49] (or writing papers with) others, and that this benefit is divided equally amongst an individual's connections. Defectors pay no cost of helping, provide no benefits, but benefit from the cooperators they are connected to. Finally, I allow the possibility that there is negative or positive synergism between cooperators, such that when two cooperators interact, their payoff is incremented by $D/(d_i d_j)$, where $D$ is the synergistic benefit. Thus, the payoff of an individual $i$ at time-step $t$, $u_i(t)$, is given by:

$$u_i(t) = \sum_{j \neq i} p_j a_{ij} \left( \frac{B}{d_j(t)} + p_i \frac{D}{d_i(t)d_j(t)} \right) - p_i C, \qquad (1)$$

where $p_j \in \{0, 1\}$ is the frequency of the cooperator allele in individual $j$, $a_{ij} = 1$ if $i$ and $j$ are connected, and 0 otherwise, and $d_j(t)$ is the degree (number of connections) of player $j$ at time $t$. An individual with payoff $u_i$ has fitness $w_i$, given by:

$$w_i = (1 + \delta)^{u_i}, \qquad (2)$$

where $\delta > 0$ is the strength of selection.

An alternative to this payoff structure can be imagined where cooperators supply a constant benefit per link. The realism of constant total or constant per link benefits depend on the absolute magnitude of benefit and the variation in degree

one observes in the networks. If variation in degree is high, as happens, for example, when social inheritance is high[36], then it is less realistic to assume individuals can ramp up production of benefits indefinitely with partner numbers such that each partner continues to enjoy the same benefit. Thus, I focus on the fixed total benefit model in the main text, and present in the Supplementary Note 2, results for the case where cooperators provide a fixed benefit to each partner and pay a fixed cost per partner.

I assume deaths occur randomly, independent of payoff or social network position. The probability of a given individual being selected to reproduce at a given time step, $\pi_i(t)$, is proportional to their fitness in the preceding time step, $w_i(t-1)$:

$$\pi_i(t) = \frac{w_i(t-1)}{\sum_j w_j(t-1)} \qquad (3)$$

At each reproduction event, the offspring copies its parent's cooperation type with probability $1-\mu$; with probability $\mu$, the offspring switches to the other type. The cooperation type of an individual remains unchanged during their lifetime.

**Evolution of linking probabilities**. To model the coevolution of the linking probabilities $p_n$ and $p_r$ with cooperation, I let them vary between individuals, and be genetically inherited from parents. With probability $\mu_l$, each of the $p'_n$ and $p'_r$ of the offspring (independently) undergo mutation, whereupon they become $p'_n = p_n + \varepsilon_n$, and $p'_r = p_r + \varepsilon_r$, where $p_n$ and $p_r$ denote the parent's linking probabilities, and $\varepsilon_n$ and $\varepsilon_r$ are distributed normally with mean zero and standard deviation $\sigma_n$ and $\sigma_r$, respectively. To restrict $p_n$ and $p_r$ to the unit interval [0,1], I set the numerical values to be at the relevant boundary if mutations fall outside this range.

To investigate how costs of making and maintaining social connections can alter the coevolutionary dynamics, I use the following extended payoff function:

$$u_i(t) = \sum_{j \neq i} p_j a_{ij} \left( \frac{B}{d_j(t)} + p_i \frac{D}{d_i(t)d_j(t)} \right) - p_i C - d_i(t)C_{\text{link}}, \qquad (4)$$

where $C_{\text{link}}$ is the per-link cost of maintaining a social connection.

**Simulations**. I analyze the above model using simulations written in the Julia programming language[50]. I initialize networks as random networks, and run the social inheritance model of Ilany and Akçay without selection for an inital burn-in period of 20 generations (i.e., $20 \times N$ time steps). This burn-in period is sufficient to produce networks that have the stationary properties of the social inheritance process[36]. Then, I allocate the cooperation trait randomly to all individuals (i.e., with expected frequency of cooperation 0.5), and turn on selection. For simulations with evolving linking probabilities, I initialize the individual $p_n$ and $p_r$ values from a normal distribution with standard deviation given by the mutational variances $\sigma_n$ and $\sigma_r$, and mean by initial values $p_n = 0.5$ (except for the fixed $p_n$ scenario) and $p_r = 0.0001$. These initial values do not have any effect on the long-run dynamics of the system as the system quickly evolves away from them.

**Code availability**. The simulation code is available at https://github.com/erolakcay/CooperationDynamicNetworks.

**Data availability**. Simulation output used in the figures is available upon request from the author.

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

## Acknowledgements

I thank J. Van Cleve, B. Morsky, and M. Smolla for comments on the manuscript, and A. Ilany for discussions. I acknowledge support from the Defense Advanced Research Projects Agency NGS2 program (Grant D17AC00005), the Army Research Office (W911NF-12-R-0012-03), the US-Israel Binational Science Foundation (2015088), and the National Academies of Science Keck Futures Initiative.

## Author contributions

E.A. conceived and designed the study, wrote the code, analyzed model, and wrote the manuscript.

## Additional information

**Competing interests:** The author declares no competing interests.

