## [Peer Review File · Nature Communications]

Reviewers' comments:

Reviewer #1 (Remarks to the Author):

This paper uses computational modeling of interacting agents on a network to study the coevolution of cooperative strategies and network formation strategies. The author confirms the well-known results that assortative networks favor the evolution of cooperation, but goes on to show that the prevalence of cooperation then selects for the broadening of network ties, which in turn destabilizes cooperation. I liked this paper. I thought the result was interesting and a valuable contribution to theory on social evolution. The literature review was also very nicely done. A number of times, I would have a concern while reading, only to have it addressed by the author within a paragraph or two. While nothing in the paper was earth shattering, it makes good a point I have not before seen made explicitly, and does so using an elegant and simple model. I think it will be of interest to both biologists and social scientists interested in cooperation (which, in principle, should be all of them). As to the paper's impact: I think the research is quite clear that reviewers are poor judges of a paper's future impact, and I am no exception. Moreover, placing too much emphasis on impact hurts the whole industry of science. So I will just repeat that I liked the paper, I learned something from it, and I'm glad I read it. I would like to see it published somewhere, and Nature Communications seems as good a place as any.

All that said, I predictably have a few concerns. I'm confident the author can address or refute all of these. Here we go.

MAJOR POINTS _____

p. 5, Equation 1: This payoff equation is quite clear to me, except for the use of 1_i in the second parenthetical term. I'm not familiar with this notation, and other readers may not be either. What does this represent?

In terms of the model description, the dynamics are well described, but the author needs to better specify it is initialized. This includes information about degree distribution. There is also a need to mention how traits are initially assigned, and how values for p_n and p_r are initially assigned.

Before going into the model results I thought the following, which perhaps the author could find useful in setting up the work.

An offspring's network will mostly look like its parents, but it will also be linked to its parent. So by positive assortment/limited dispersal, this will increase cooperation. As cooperation spreads, making more connections becomes valuable, leading to a highly connected network. This provides strong incentives to cheat, since everyone is sharing with everyone. There is a literature on dispersal and mobility showing that cooperators are best served by limited dispersal and restricted mobility, while defectors are best served by wide-ranging dispersal and mobility. This may help to explain (to the readers – I'm sure the author gets this) why there is this shift from selection for cooperation to selection to defection, since the

network structure acts to move from one system to the other. Some example references:

Dispersal:

Koella JC (2000) The spatial spread of altruism versus the evolutionary response of egoists. *Proc R Soc Lond B* 267:1979-1985.

Mobility:

Smaldino PE, Schank JC (2012) Movement patterns, social dynamics, and the evolution of cooperation. *Theor Popul Biol* 82: 48-58.

p. 7, lines 143-144/Figure 1. Why does cooperation break down at very high levels of p_n ? It's not clear to me why this should be the case.

Results: "Fixed link probabilities"

In Figure 2, the negative slope of the cooperation threshold in these graphs indicates that there is some idea number of ties, with parental ties are worth more. The synergy basically gives cooperators an advantage relative to exploitative defectors. It would be useful to think about how this works in terms of well-known cooperative dilemmas such as the prisoner's dilemma and the snowdrift game, and if this dynamic possibly illustrates something that CANNOT be captured by modeling those games more explicitly. The author actually does some of the analysis in the supplement, which I very much appreciate. What I'd like is a bit more discussion of the implications, which at present are mentioned briefly but not explored. More generally, I think the results in the paragraph "Fixed link probabilities" warrant a bit more explanation. Since this paper presents a model and not empirical research, the added value is in the depth of analysis you can give to explaining your simulation results. I think the results are really interesting. I want to know exactly why they emerge.

Why were the simulations in figures 1 and 2 only run for 500 generations, when the simulations in figure 4 were run for 10^5 generations and the simulations in Figure 3 appear to be run for $5 \cdot 10^4$ time steps (is this then 500 generations again?). Also, the caption in Fig 3 is confusing because it notes that cooperation starts to collapse around generation 100 – is this then $time = 10^4$?

Results: "Costs of linking can rescue cooperation"

For high levels of C_{link} , intermediate values of B seem to yields the highest long-term rates of cooperation. Why? This is interesting but demands an explanation. It appears related to the fact that p_n is minimal at the maximal values of cooperator frequency. So the agents aren't forming links with their parents' links. In this case, agents are mostly just linking to their parents, and that scenario makes sense why it would lead to maximal cooperation. You might note this. It was initially not clear to me why this network strategy would be selected for at all. Your first explanation, that p_r is evolving to higher values, doesn't explain the pattern we see in p_n , which is what I believe is really driving the result. Your explanation about parent-offspring conflict (p. 10) makes a lot of sense and is very cool. You might highlight this more. In fact, I'd like to see some analysis looking more directly at parent-offspring conflict. If you could find some measure of parent- vs. offspring-level selection for p_n , for example, it would be really interesting.

For the case with synergistic cooperation and link costs, you eventually reach a point where no links other than the parent link are formed. This makes sense from the model standpoint, but you might comment on the ecological implications, since it doesn't seem realistic for a social species.

Discussion, p. 12, lines 220-221: "I use a simple dynamical network model that is able to reproduce important characteristics of animal social structure based on social inheritance [33]"

The model you cite did not have selection on cooperation-based payoffs. Since the network structure in the present model results in part from that selection, that claim must at least be clarified, and may in fact not hold at all.

This is further discussed at the bottom of p. 15 – parameters in Ref 33 are directly linked to the present model. I think the author needs to show that the network properties (e.g. clustering) for the present model are consistent with the network properties the model in Ref 33, as there are important differences that could affect network structure. Alternatively, just remove the claim about the ecological realism of the network.

MINOR POINTS_____

The model and results were reminiscent of an earlier paper by Smaldino et al. They get a sort of complementary result: conditions that initially favor defection give rise to conditions that favor cooperation. It might be worth drawing this connection.

Smaldino PE, Schank JC, McElreath R (2013) Increased costs of cooperation help cooperators in the long run. *American Naturalist* 181:451-463.

Line 2: "Cooperation is easy to evolve." This was extremely refreshing to see. So many papers begin with the great mystery that is cooperation, but the truth is that we understand the mechanisms by which it can evolve very well.

p. 4, line 68: should be "ARE selected to increase"

p. 7, line 150: "the coauthor game..." I think if you're going to use this terminology, you should briefly explain why this game should evoke analogies to coauthorship networks, since not all readers will be familiar with Jackson and Wolinsky's paper.

p. 9, line 183: You might cite Rosenzweig 1971 for "paradox of enrichment," since readers who aren't evolutionary ecologists may not know the reference. Alternatively, if the effect you get occurs for different reasons than Rosezweig's, you might scrap the term to avoid confusion.

p. 13, lines 241-242: "I show that the structure of the society that favors cooperation can itself fall victim to cooperation."

It is well known that limited dispersal and positive assortment lead to cooperation, and that increased mixing hurts cooperation. The author has shown that cooperation evolving through limited dispersal can itself select for increased mixing, destroying cooperation. This

is the key insight of the paper, and it's a valuable one. But it only becomes a problem when ties are not costly, and this is true in few social organisms. In the case of humans, at the most extreme, countless institutions have arisen to mediate cooperation in larger groups. On the other hand, consider that the internet has dramatically decreased the cost of ties, and look what's happened. There's probably a lot of things to be said about this, perhaps more than this one paper can support.

Signed:
Paul Smaldino

Reviewer #2 (Remarks to the Author):

The study by Akçay investigates the coevolutionary dynamics on a dynamical network structure of game strategies and linking probabilities of individuals playing according to the rules of a non-cooperative game. The considered game is the coauthor game, firstly introduced by Jackson and Wolinsky (1996), although supplementary materials also provide results for the more classical Prisoner's Dilemma game. Individuals on the network are engaged in pairwise interactions with all their neighbors accumulating payoffs from them. A standard birth-death process is modeled where deaths occur with random probability while most fitted individuals have higher probability to reproduce offsprings when an individual is replaced. The link formation model for incoming individuals is principally based on a recent work (Ilany and Akçay, Nat Comm 2016) where the authors introduce a novel dynamical process to obtain similar network structures of those present in animal societies. Here, the game dynamics is introduced investigating the evolution of cooperation by numerical simulations.

Although the topic is really hot and the model can represent a significant contribution to the existing literature, I have too many caveats regarding the robustness of the presented results and on how the current study has been performed to support it for publication. Several explanations and discussions are omitted whereas the numerical simulation setting/analysis is not particularly well-conducted. Moreover, no empirical data to be compared with the model results are presented. Overall, the study is very interesting but, at its current state, it does not match the standards of novelty, results accuracy and discussion, to justify a publication in Nature Communications. I would suggest the author to resubmit his work once the comments below are addressed.

Major comments:

- Model parameters: three main features of the model are investigated, i.e., (1) the evolution of linking probabilities, (2) the introduction of synergistic benefit for cooperators, (3) the influence of linking cost, among others (strong/weak selection, game payoffs, mutation rates). However, not all of them are separately analyzed and satisfactorily discussed. I suggest the author to focus on only a couple of them before studying the three of them together without a good discussion of the model parameters and their calibration. Numerical simulations are useful to cover all (or more) parameter values and their influence

on the model. In general, too many model parameters at the same time are considered leading to a very difficult interpretation of the results.

- Network structure analysis: although the network size, i.e. $N=100$, can perhaps be considered fine for small social animal communities, no hint on results for other network sizes is given. More importantly, a detailed analysis on the evolution of the average degree, and of degree distribution of the final networks, is totally missing. One possible explanation of the results can be that the average degree is boundlessly increasing reaching the almost well-mixed population scenario, which usually favors defectors.

- Results convergence: according to figure captions, 500 generations are simulated reporting results of the averages of the last 400 generations. Although 500 generations are enough for a small network of 100 nodes to usually get convergence (5x), averaging over the 80% of the simulation time can dramatically affect the reported results. In fact, in order to measure final network statistics, it is more accurate to let the population evolve for 400 generations and then averaging over the last 100 generations, for instance. This allows to better understand the converge, if any, to a cooperation/defection equilibrium and to present results with more accuracy on the final population state. Overall, considering this methodology, all the results seem affected by a huge amount of noise.

- Utility functions: no discussion is provided to justify the choice of utility functions in Eq. (1) and (4). While the literature review in the Introduction is well-conducted, no related references are given in the Model section. Furthermore, it is not clear why a +1 is added in Eq. (4) with respect to (1), nor why the synergistic benefit is defined multiplying degrees instead than summing them at the denominator, for instance, or using another possible function. Is there any biological explanation in introducing this synergistic benefit only to cooperators and not for defectors interacting with cooperators? Clarifications required.

Minor comments:

- In general, there are quite some convoluted/unclear sentences in the manuscript, which I will not exhaustively list. For resubmission, I would advise to have a native English speaker look over the manuscript once more. The abstract, in particular, can be clearer.

- The Simulations section at line 133 can go to the Supplementary Material instead.

- The results on PD games, instead than those for the coauthor games, are usually more frequent in the numerical simulation literature, or is there any particular biological explanation to present in the main text the coauthor game instead than the PD?

- Figure 3 results can also show the average values in order to better understand a pattern in Fig. 3(a).

- In order to avoid too many parameters, only weak selection results can be presented. Strong selection can be very biased having such small network sizes.

Reviewer #3 (Remarks to the Author):

This paper studies coevolution of cooperation and network structure in a game theoretical model. The main finding of the paper is as follows. In what is called a "coauthor game", cooperation is favored by natural selection when the probability of random linking, p_r , is low. However, when this linking probability itself can evolve, it evolves towards a larger value. This creates a negative feedback and cooperation eventually collapses. The author also finds that this collapse is rescued by linking cost or synergistic benefits of cooperation.

I enjoyed reading the paper. In fact, the paper is rich with theoretical implications, and the mechanisms of collapse and rescue of cooperation presented here are novel. I have several suggestions to improve the paper, as described below.

[1] Reference to previous works on evolution of cooperation in a dynamic-network setting is unfortunately not rich enough (only citations 30-32). In particular, many studies have intensively investigated the effect of dynamic linking (or dynamic link-weight adjustment) on evolution of cooperation. Those works typically assume that the link is maintained (or the link weight is increased) when one benefits from the interaction with the partner, and otherwise the link is broken (or the link weight is decreased). To list a few, Huang, Zheng & Yang (2015; Scientific Reports), Fu, Hauert, Nowak, & Wang (2008;PRE), and Skyrms and Pemantle (2000; PNAS). Consider citing those (and other) papers.

[2] The author finds that the effect of p_n is quite marginal (page 7). However, I naively expect that inheriting links from one's parent, especially when cooperation is prevalent in the population, should be very beneficial, because it is highly likely that his/her parent would have many cooperative neighbors. Please provide more explanations to that.

[3] Model (page 4): Because the author's model considers probabilities of link-inheritance and random-connection PER INDIVIDUAL, the absolute number of connections increases with increased population size, N . This makes me wonder whether the author's result is scale-free or not, because many previous studies have shown the importance of absolute neighborhood size (see, for example, reference 27). Put differently, I wonder if the result is qualitatively unchanged if N becomes two/five/ten times larger, or so. I naively expect that this would increase the neighborhood size and would considerably disfavor cooperation. Is that right?

[4] The rate of strategy evolution is controlled by δ , whereas the rate of linking probability evolution is controlled by μ_l and σ 's in this paper. I wonder if changing their relative balance could change the results. In particular, can we observe a cyclic behavior of p_r increasing, cooperation collapsing, followed by the decrease of p_r , and by re-emergence of cooperation? Or is evolution always in one way, in the sense that, once the increased level of p_r undermines cooperation, cooperation never recovers evolutionarily?

[5] " 1_i " in eqs.(1) and (4) should be " p_i ".

[6] "work in exactly the same way" (lines 467-468 in SI) is ambiguous. I think the author wants to point out that both the C-term and the C_link-term are proportional to the number of connections, $d_i(t)$ in eq.(SI-1) whereas it is not the case in eq.(4) in the main text. Please add more words here.

[7] Clarify parameters used in each subsection: in the "Fixed linking probabilities" section I guess $C_{\text{link}} = 0$. In the "Coevolution of linking probabilities ..." subsection I guess $C_{\text{link}} = D = 0$.

[8] I occasionally find minor grammatical errors. For example, "the higher linking costs have to be [to] maintain it" (abstract), "..., mostly independent[ly] of the probability" (page 4, top). Please review the whole manuscript again and clean them off.

Response to reviewers

For: “Collapse and rescue of cooperation in evolving dynamic networks” Erol Akcay, submitted to Nature Communications

I thank the three reviewers for their insightful comments and helpful suggestions. I have made numerous changes to the manuscript following their suggestions, which I believe improved the content and presentation. Below, I respond to reviewer comments in detail, explaining changes made in response.

Reviewer #1 (Remarks to the Author):

This paper uses computational modeling of interacting agents on a network to study the coevolution of cooperative strategies and network formation strategies. The author confirms the well-known results that assortative networks favor the evolution of cooperation, but goes on to show that the prevalence of cooperation then selects for the broadening of network ties, which in turn destabilizes cooperation. I liked this paper. I thought the result was interesting and a valuable contribution to theory on social evolution. The literature review was also very nicely done. A number of times, I would have a concern while reading, only to have it addressed by the author within a paragraph or two. While nothing in the paper was earth shattering, it makes good a point I have not before seen made explicitly, and does so using an elegant and simple model. I think it will be of interest to both biologists and social scientists interested in cooperation (which, in principle, should be all of them). As to the paper’s impact: I think the research is quite clear that reviewers are poor judges of a paper’s future impact, and I am no exception. Moreover, placing too much emphasis on impact hurts the whole industry of science. So I will just repeat that I liked the paper, I learned something from it, and I’m glad I read it. I would like to see it published somewhere, and Nature Communications seems as good a place as any.

I thank Dr. Smaldino for these encouraging comments and am glad that he finds the contribution valuable.

Major points

p. 5, Equation 1: This payoff equation is quite clear to me, except for the use of 1_i in the second parenthetical term. I’m not familiar with this notation, and other readers may not be either. What does this represent?

I thank Dr. Smaldino (and Reviewer 3) for catching this holdover notation from a previous version: 1_i was the indicator variable for cooperation. In the most recent notation, it should read p_i , the frequency of cooperation in individual i .

In terms of the model description, the dynamics are well described, but the author needs to better specify it is initialized. This includes information about degree distribution. There is also a need to mention how traits are initially assigned, and how values for p_n and p_r are initially assigned.

I now provide a more detailed description of the implementation and initialization of the simulations. Briefly, networks are initialized as random networks, and run without selection for an initial burn-in period of 20 generations (i.e., 20 x (network size) time steps). This burn-in period is sufficient to produce networks that have the stationary properties of the social inheritance process (Ilany & Akcay 2016). Then, I allocate the cooperation trait randomly to all individuals, and turn on selection. In other words, the expected starting frequency of cooperation is 0.5. For simulations with evolving p_n and p_r , I initialize the individual p_n and p_r values from a normal distribution with standard deviation given by the mutational variances σ_n and σ_r , and mean by initial values $p_n = 0.5$ and $p_r = 0.0001$. These initial values do not have any effect on the long-run dynamics of the system as the system quickly evolves away from them.

Before going into the model results I thought the following, which perhaps the author could find useful in setting up the work. An offspring's network will mostly look like its parents, but it will also be linked to its parent. So by positive assortment/limited dispersal, this will increase cooperation. As cooperation spreads, making more connections becomes valuable, leading to a highly connected network. This provides strong incentives to cheat, since everyone is sharing with everyone. There is a literature on dispersal and mobility showing that cooperators are best served by limited dispersal and restricted mobility, while defectors are best served by wide-ranging dispersal and mobility. This may help to explain (to the readers – I'm sure the author gets this) why there is this shift from selection for cooperation to selection to defection, since the network structure acts to move from one system to the other. Some example references: Dispersal: Koella JC (2000) The spatial spread of altruism versus the evolutionary response of egoists. *Proc R Soc Lond B* 267:1979-1985. Mobility: Smaldino PE, Schank JC (2012) Movement patterns, social dynamics, and the evolution of cooperation. *Theor Popul Biol* 82: 48-58.

I thank Dr. Smaldino for these comments: indeed there is a connection between random linking and social inheritance on one hand and dispersal and philopatry on the other. The new results with larger networks I present in this version are also directly relevant to this connection. In larger networks, persistent (but not

stable) polymorphisms in cooperation can arise, and cooperators and defectors evolve different linking strategies. I now elucidate this connection further in the discussion, using the citations suggested by Dr. Smaldino among others.

p. 7, lines 143-144/Figure 1. Why does cooperation break down at very high levels of p_n ? It's not clear to me why this should be the case.

Dr. Smaldino is right to point out that the first main result probably required more explaining. The finding that cooperation requires little random linking is essentially a classical case of kin selection: for a given level of social inheritance, random linking decreases the relatedness between individuals (see the new Supplementary Figure SI 1), and hence cooperators are less likely to assort with each other. The fact that p_n has relatively little effect for most of its range, except for very high p_n which also works against cooperation is due to the connectedness of the population getting too high, which also reduces the amount of assortment that one can have (see Figure 5 in Ilany & Akcay, 2016). I have now included this explanation in the main text when I talk about Figure 1.

Results: “Fixed link probabilities” In Figure 2, the negative slope of the cooperation threshold in these graphs indicates that there is some idea number of ties, with parental ties are worth more. The synergy basically gives cooperators an advantage relative to exploitative defectors. It would be useful to think about how this works in terms of well-known cooperative dilemmas such as the prisoner's dilemma and the snowdrift game, and if this dynamic possibly illustrates something that CANNOT be captured by modeling those games more explicitly. The author actually does some of the analysis in the supplement, which I very much appreciate. What I'd like is a bit more discussion of the implications, which at present are mentioned briefly but not explored. More generally, I think the results in the paragraph “Fixed link probabilities” warrant a bit more explanation. Since this paper presents a model and not empirical research, the added value is in the depth of analysis you can give to explaining your simulation results. I think the results are really interesting. I want to know exactly why they emerge.

Dr. Smaldino's interpretation is correct that synergy gives an advantage to cooperators that the defectors cannot take advantage of. In this sense, there is not really that much different going on here than more common (non-additive) prisoner's dilemma game. The main difference between the model I discuss in the main text and the “vanilla” PD game is the difference in how benefits (synergistic or additive) are handled as a function of degree, which is a non-trivial issue in heterogenous networks without uniform degree throughout. I expanded, in the model description section, the comparison between the “coauthor” and “PD” models, and my justification for focusing on the coauthor model in the main text. As mentioned above, I also expanded my explanation of why the fixed linking probability results come about, which are mainly in line with classical

intuitions, with the addition of the dilution of benefits effect.

Why were the simulations in figures 1 and 2 only run for 500 generations, when the simulations in figure 4 were run for 10^5 generations and the simulations in Figure 3 appear to be run for $5 * 10^4$ time steps (is this then 500 generations again?).

The main difference between Figures 1/2 and 4 is that the latter let the linking probabilities evolve, which gives a much bigger state space to explore (especially since the linking traits are quantitative, and evolve by small effect mutations). Therefore, to let the system sufficiently explore both the space of linking properties and cooperation to get the average behavior, a longer time-period is needed for the latter simulations. I added a note to the caption of Figure 4 to this effect.

Figure 3 on the other-hand is meant to illustrate the dynamics created by the negative feedback (rather than long-term behavior of the system), and focuses on the initial 3000 generations for that reason.

Also, the caption in Fig 3 is confusing because it notes that cooperation starts to collapse around generation 100 – is this then time = 10^4 ?

That’s correct. In this version, I replaced Figure 3 with the weak-selection version, as explained above, and the confusing description is not in the caption anymore.

Results: “Costs of linking can rescue cooperation” For high levels of C_{link} , intermediate values of B seem to yields the highest long-term rates of cooperation. Why? This is interesting but demands an explanation. It appears related to the fact that p_n is minimal at the maximal values of cooperator frequency. So the agents aren’t forming links with their parents’ links. In this case, agents are mostly just linking to their parents, and that scenario makes sense why it would lead to maximal cooperation. You might note this. It was initially not clear to me why this network strategy would be selected for at all. Your first explanation, that p_r is evolving to higher values, doesn’t explain the pattern we see in p_n , which is what I believe is really driving the result. Your explanation about parent-offspring conflict (p. 10) makes a lot of sense and is very cool. You might highlight this more. In fact, I’d like to see some analysis looking more directly at parent- vs. offspring-level selection for p_n , for example, it would be really interesting.

The pattern of p_n at strong selection (upper row of Figure 4) is indeed interesting, though I don’t think it is driving the non-monotonicity in B, for two reasons: first, p_n doesn’t seem to experience much selection one way or another under weak selection (lower row of Figure 4), yet we obtain the same non-monotonic

pattern in B where cooperation is highest at intermediate values of B. Second, the fixed linking probability results show that p_n in general has very weak influence on cooperation, as long as it's not too high. So, I am confident that the pattern of average cooperation is driven mostly by p_r .

Rather, the variation in p_n seems to be a consequence and not a cause of the patterns in cooperation and p_r . As I tried to explain on lines 193-208 on the original manuscript, in the coauthor game for cooperative parents, there is a parent-offspring conflict over linking: the parents would like offspring to only link to themselves, and not anyone else, so as to ensure the benefits they receive from their cooperative offspring are not diluted. With strong selection, an individual who receives exclusive benefits from a couple of connections has a highly disproportionate fitness (probability of reproducing again), which locks in this fitness benefit further (as more offspring get singly connected to this one parent), until this central node dies (which happens randomly in the model). When p_r is also low, lineages that have low p_n produce these "super-reproducers" with higher frequency, and therefore get selected. However, when p_r is higher (which evolves regardless of p_n in cooperative populations when the cost of linking is low relative to the benefit from cooperation), offspring will already be not exclusively connected to their parents, with one or more random connections, and therefore the fitness lost by parents by the dilution (which is proportional to $1/\text{degree}$) is less. In addition, having higher p_n in this case re-allocates benefits from random individuals to parent's connections, which tend to be from the same lineage as the offspring (and therefore also cooperative). These explain why higher p_r also selects for higher p_n . With weak selection, the benefit to receiving exclusive benefits is much less disproportionate, and the benefits from connecting to other relatives compensates for the loss of exclusive benefits to the parent.

I was somewhat unsure about what to do with the parent-offspring conflict aspect in strong selection, since it becomes most pronounced under an arguably edge-condition of strong selection, which creates a very strong skew as a function of connection traits (as explained above), and is somewhat tangential to the overall message of the paper. This is why even in the initial submission I had vacillated about including them in the main text. In this version, in part because I added new analyses that are more central to the story, I decided to relegate the strong selection results to the supplement. Note that in the initial submission, the fixed linking probabilities results shown were also under strong selection, though they are qualitatively robust to strength of selection, so I also replaced them with weak-selection (with strong selection simulations in the supplement).

For the case with synergistic cooperation and link costs, you eventually reach a point where no links other than the parent link are formed. This makes sense from the model standpoint, but you might comment on the ecological implications, since it doesn't seem realistic for a social species.

I now comment on this pattern when discussing the results with synergism and

in the discussion. Briefly, I agree with Dr. Smaldino that the conditions where cooperation is maintained with synergism are not realistic from the perspective of what we know of social structure in nature. One potential answer to this dilemma is found in the results with fixed p_n : when high social inheritance is exogenously maintained (which is a possibility since there is relatively little selection on it due to co-evolution with cooperation), we can get cooperative populations with realistic network structure (like those found by Ilany and Akcay, 2016).

Discussion, p. 12, lines 220-221: “I use a simple dynamical network model that is able to reproduce important characteristics of animal social structure based on social inheritance [33]” The model you cite did not have selection on cooperation-based payoffs. Since the network structure in the present model results in part from that selection, that claim must at least be clarified, and may in fact not hold at all. This is further discussed at the bottom of p. 15 – parameters in Ref 33 are directly linked to the present model. I think the author needs to show that the network properties (e.g. clustering) for the present model are consistent with the network properties the model in Ref 33, as there are important differences that could affect network structure. Alternatively, just remove the claim about the ecological realism of the network.

Dr. Smaldino is right that under strong selection, properties of networks that arise under selection can be different than neutral networks with the same p_n and p_r values, as I previously remarked in the discussion as well as the supplementary material. However, this issue does not arise under weak selection (and $\delta=0.1$ is weak enough for these purposes). I agree that arguably (see below) some of the strong selection results are biologically suspect. That is why I focused on weak selection results in this version, which show the same self-limiting properties of cooperation, while retaining the properties (including ecological realism) of the neutral networks as presented by Ilany and Akcay.

To go into more detail, as explained above, the strong-selection regime tends to create star-networks with many offspring connected to a parental node (until the node dies and breaks the star), which is certainly less realistic. That is one reason I do not regard the strong selection results as especially realistic, but had decided to present them for the sake of completeness (and because they illustrate an interesting conceptual point –in an exaggerated fashion– the potential for parent-offspring conflict over linking). The strong selection results are now relegated to the supplement.

For weak selection, differences in mean degree and clustering between networks under neutrality and *weak selection* is relatively small, as can be seen in the current Supplementary Figure SI 8 (previously Supplementary Figure SI2), where real p_n of non-neutrally evolving networks are compared to estimates of p_n calculated using the neutral model approximation. These approximations are based on the mean degree and clustering of the neutral networks, so the fact we can

recover true linking probabilities accurately means that these mean properties of networks evolving under weak selection are a good match to their neutral counterparts.

Another aspect of the ecological realism question is the current model does not necessarily predict high rates of social inheritance that we found in Ilany and Akcay 2016 to evolve. However, there is relatively little tension here, since at least with weak selection, there also isn't much selection for p_n to increase or decrease. This suggests that if other factors that are not modeled here (e.g., the need for social learning or support in social conflicts) favor higher p_n , selection for cooperation as in my model will not interfere with the evolution of higher p_n . Furthermore, as the new results I include show, if one of these exogenous factors maintains social inheritance at high levels, it can promote high levels cooperation and high average payoffs. That raises the interesting (but not modeled here) possibility that high p_n can be selected for by between group selection, as groups with traits causing high social inheritance would have higher productivity, and this would not be strongly opposed by within-group selection on p_n (which is weak or non-existent). I now make this point in the discussion.

Minor points

The model and results were reminiscent of an earlier paper by Smaldino et al. They get a sort of complementary result: conditions that initially favor defection give rise to conditions that favor cooperation. It might be worth drawing this connection. Smaldino PE, Schank JC, McElreath R (2013) Increased costs of cooperation help cooperators in the long run. *American Naturalist* 181:451-463.

I thank Dr. Smaldino for this reference, which I had previously missed. It is indeed quite relevant, and I discuss the connection in the discussion in the subsection on parallels with spatially structured population models. As I elaborate there, I find that there are both some parallels but also some important differences between the two papers.

Line 2: "Cooperation is easy to evolve." This was extremely refreshing to see. So many papers begin with the great mystery that is cooperation, but the truth is that we understand the mechanisms by which it can evolve very well.

Thank you, I wholeheartedly agree!

p. 4, line 68: should be "ARE selected to increase"

Thank you, fixed.

p. 7, line 150: "the coauthor game..." I think if you're going to use this terminology, you should briefly explain why this game should evoke analogies to coauthorship networks, since not all readers will be familiar with Jackson and Wolinsky's paper.

I don't use the coauthor game terminology any other place in the main text (only in the SI when I talk about the PD game), and I have added a more explicit motivation of the payoff function in the model description that hopefully makes it clearer. The name, as far as I can tell, has no necessary relationship to the real coauthorship networks; rather it seems to me to be an inside joke about the efficiency of working on too many papers with different coauthors at a time.

p. 9, line 183: You might cite Rosenzweig 1971 for “paradox of enrichment,” since readers who aren't evolutionary ecologists may not know the reference. Alternatively, if the effect you get occurs for different reasons than Rosezweig's, you might scrap the term to avoid confusion.

I had hesitated about citing Rosenzweig, since the phenomenon has no real mechanistic relationship to what happens in predator-prey dynamics. I did keep the phrase (used only once), since it intuitively captures the phenomenon we observe. I decided to add a note saying that it is unrelated to Rosenzweig's paradox of enrichment by way of avoiding any inadvertent confusion.

p. 13, lines 241-242: “I show that the structure of the society that favors cooperation can itself fall victim to cooperation.” It is well known that limited dispersal and positive assortment lead to cooperation, and that increased mixing hurts cooperation. The author has shown that cooperation evolving through limited dispersal can itself select for increased mixing, destroying cooperation. This is the key insight of the paper, and it's a valuable one. But it only becomes a problem when ties are not costly, and this is true in few social organisms. In the case of humans, at the most extreme, countless institutions have arisen to mediate cooperation in larger groups. On the other hand, consider that the internet has dramatically decreased the cost of ties, and look what's happened. There's probably a lot of things to be said about this, perhaps more than this one paper can support.

I agree with these comments, especially with the conclusion that there is a lot to be said about this that go beyond the scope of this paper. By way of pointing in some of these directions, I discuss the evolution of costs of linking in the discussion, pointing out that while they probably would not evolve individually (as they provide no direct benefits), they might evolve through group selection processes, but only in conjunction with social inheritance, as the new results I added show. I now expanded the discussion of this issue at the beginning of the discussion section.

Signed: Paul Smaldino

Reviewer #2 (Remarks to the Author):

The study by Akçay investigates the coevolutionary dynamics on a dynamical network structure of game strategies and linking probabilities of individuals playing according to the rules of a non-cooperative game. The considered game is the coauthor game, firstly introduced by Jackson and Wolinsky (1996), although supplementary materials also provide results for the more classical Prisoner's Dilemma game. Individuals on the network are engaged in pairwise interactions with all their neighbors accumulating pay-offs from them. A standard birth-death process is modeled where deaths occur with random probability while most fitted individuals have higher probability to reproduce offsprings when an individual is replaced. The link formation model for incoming individuals is principally based on a recent work (Ilany and Akçay, Nat Comm 2016) where the authors introduce a novel dynamical process to obtain similar network structures of those present in animal societies. Here, the game dynamics is introduced investigating the evolution of cooperation by numerical simulations.

Although the topic is really hot and the model can represent a significant contribution to the existing literature, I have too many caveats regarding the robustness of the presented results and on how the current study has been performed to support it for publication. Several explanations and discussions are omitted whereas the numerical simulation setting/analysis is not particularly well-conducted. Moreover, no empirical data to be compared with the model results are presented. Overall, the study is very interesting but, at its current state, it does not match the standards of novelty, results accuracy and discussion, to justify a publication in Nature Communications. I would suggest the author to resubmit his work once the comments below are addressed.

Major comments:

- Model parameters: three main features of the model are investigated, i.e., (1) the evolution of linking probabilities, (2) the introduction of synergistic benefit for cooperators, (3) the influence of linking cost, among others (strong/weak selection, game payoffs, mutation rates). However, not all of them are separately analyzed and satisfactorily discussed. I suggest the author to focus on only a couple of them before studying the three of them together without a good discussion of the model parameters and their calibration. Numerical simulations are useful to cover all (or more) parameter values and their influence on the model. In general, too many model parameters at

the same time are considered leading to a very difficult interpretation of the results.

I thank the reviewer for the suggestion, though I admit to being somewhat unsure about how to follow it. In particular, I feel that my original analysis as well as the current one did not focus on an unduly large number of parameters. Beyond the regular payoff parameters (which are unavoidable in a social evolution model), I mainly focus on the effects of three main variables: the linking probabilities, and the cost of linking. My most novel main results come from letting the linking probabilities co-evolve with cooperation, which of course adds more complexity than just letting cooperation evolve, but that is precisely the purpose of the model. Other parameters (such as strength of selection, mutation rate, population size) are auxiliary parameters that are also unavoidable in any selection model, and my results are robust to changes in them.

That said, in part in response to comments by the other reviewers, I did undertake some major changes that might also address the reviewer's concern. In particular, I now moved all the strong selection results to the appendix (as explained above in response to Reviewer 1), which simplifies the presentation of the results. I also have a new subsection where I keep the social inheritance parameter p_n fixed and look at the co-evolution of random linking probability with cooperation only, which reveals some new interesting results.

- Network structure analysis: although the network size, i.e. $N=100$, can perhaps be considered fine for small social animal communities, no hint on results for other network sizes is given.

The reviewer is correct to point out that I have not included any results with network sizes other than 100. This was mainly because the results are in general not sensitive to network size, but it is completely fair to demand these sensitivity analysis to network size. I now present results with network sizes of $N=200$ and $N=500$ in the supplementary material, which show that the population mean patterns are not sensitive to network size. The only effect of network size is that it changes the relevant scale of p_r , the random linking probability, since it is defined as a per-capita probability. For example, the same p_r value will yield (roughly) twice as many expected number of random connections in a network of size 200 compared to 100.

More importantly, a detailed analysis on the evolution of the average degree, and of degree distribution of the final networks, is totally missing. One possible explanation of the results can be that the average degree is boundlessly increasing reaching the almost well-mixed population scenario, which usually favors defectors.

It is easy to show that cooperation is not directly tied to the average degree of the networks. For example, networks with $p_r=0.01$ (say, in Figure 1, with $B=2$) and $p_n=0.2, 0.6$, and 0.8 , will all have high levels of average cooperation, but have vastly different average degrees. I now point this out in the discussion

and have also included a supplementary figure to show this point (SI Figure 2, where the same frequency of cooperation can be obtained with a wide range of degrees). It is true that results in static networks identify average degree as a determinant of whether cooperation evolves or not. But this finding does not generalize to dynamic networks evolving according to social inheritance. In both cases, the operative variable is the assortment (or relatedness) rather than degree, and in dynamic networks with social inheritance, there is only a loose relation between degree and assortment.

- Results convergence: according to figure captions, 500 generations are simulated reporting results of the averages of the last 400 generations. Although 500 generations are enough for a small network of 100 nodes to usually get convergence (5x), averaging over the 80% of the simulation time can dramatically affect the reported results. In fact, in order to measure final network statistics, it is more accurate to let the population evolve for 400 generations and then averaging over the last 100 generations, for instance. This allows to better understand the converge, if any, to a cooperation/defection equilibrium and to present results with more accuracy on the final population state. Overall, considering this methodology, all the results seem affected by a huge amount of noise.

To clarify, “generation” in the paper refers to N deaths and births, where N is the network size (as was mentioned in the captions) so 500 generations with $N=100$ is 50000 birth and death events. In inspections of simulations, convergence to stationary network properties happens in < 20 generations (2000 time steps with $N=100$) under neutrality, and with selection, cooperation quickly approaches high or low levels within a similar time frame (as can be seen, for example, in Figure 3). Therefore, there is no reason to believe that the averaging procedure introduces excess noise – or likely closer to the reviewer’s meaning, a bias due to initial conditions and transient dynamics. As an illustration, below, I include a figure depicting the same simulations as Figure 1 of the main text, except that I discard the first 400 generations and only average over the last 100 generations (i.e., the last 10000 time steps). As the figure shows, there is virtually no difference, showing the averaging does not introduce undue bias.

- Utility functions: no discussion is provided to justify the choice of utility functions in Eq. (1) and (4). While the literature review in the Introduction is well-conducted, no related references are given in the Model section. Furthermore, it is not clear why a +1 is added in Eq. (4) with respect to (1), nor why the synergistic benefit is defined multiplying degrees instead than summing them at the denominator, for instance, or using another possible function. Is there any biological explanation in introducing this synergistic benefit only to cooperators and not for defectors interacting with cooperators? Clarifications required.

I thank the reviewer for this comment. I added a more explicit motivation for the payoff functions I used to the model section, which I believe capture fairly intuitive trade-offs.

The +1 was a typo; removed in the current edition – thank you for catching it.

As for the synergistic benefits, this form seems the most intuitive extension of the coauthor payoff function to include synergism (which was also used by Jackson and Wolinsky). For both the additive and non-additive benefits in the coauthor game, $1/d_i$ represents individual i 's investment to a single partner. In social evolution additive benefits are generally taken as proportional to this investment while multiplicative or synergistic benefits proportional to the product of investments, which yields $1/(d_i d_j)$. It is also well-known (as I discuss in the Discussion) that synergistic benefits arise naturally from wide-spread mechanisms such as reciprocity.

Minor comments:

- In general, there are quite some convoluted/unclear sentences in the manuscript, which I will not exhaustively list. For resubmission, I would advise to have a native English speaker look over the manuscript once more. The abstract, in particular, can be clearer.

I am exceedingly grateful for this advice.

- The Simulations section at line 133 can go to the Supplementary Material instead.

Thank you for the suggestion. I decided to keep the short description of the simulation procedure together with the rest of the model section and the link to the code accessible without downloading a supplementary material first.

- The results on PD games, instead than those for the coauthor games, are usually more frequent in the numerical simulation literature, or is there any particular biological explanation to

present in the main text the coauthor game instead than the PD?

This issue is related to the motivation of the payoff functions above. I chose to focus on the coauthor game in the main text, since I believe it is the more relevant function for complex networks with varying number of connections. This is because it captures the idea that cooperative individuals will have a finite amount of resources (or time) they can invest in others, and with higher degree, the investment necessarily has to be divided up between more partners. In contrast, the PD game where each connection receives a constant benefit from a node implies that a cooperator can produce benefits regardless of the total magnitude of investment required. I believe the coauthor game to be a more realistic state of affairs, and present the PD game for completeness' sake. It is of course possible to imagine intermediate cases where the total benefits given out (and total costs) increase non-linearly (e.g. in a saturating way for the benefits) with the degree of a cooperator. The behavior of such a model would depend on the shape of the benefit function but for most reasonable shapes would fall in between the coauthor and PD game cases. As mentioned above, I made these points more explicit in the motivation of the payoff function.

- Figure 3 results can also show the average values in order to better understand a pattern in Fig. 3(a).

I thank the reviewer for this helpful suggestion, which I followed.

- In order to avoid too many parameters, only weak selection results can be presented. Strong selection can be very biased having such small network sizes.

I agree with the reviewer's comment (as also explained above), and decided to relegate the strong selection results to the SI, where they are presented for completeness' sake. Although the main patterns remain unchanged between weak and strong selection, strong selection does cause some extreme patterns in the network structure (see response to Reviewer 1 above) that are perhaps biologically not realistic.

Reviewer #3 (Remarks to the Author):

This paper studies coevolution of cooperation and network structure in a game theoretical model. The main finding of the paper is as follows. In what is called a "coauthor game", cooperation is favored by natural selection when the probability of random linking, p_r , is low. However, when this linking probability itself can evolve, it evolves towards a larger value. This creates a negative feedback and cooperation eventually collapses. The author also finds that this collapse is rescued by linking cost or synergistic benefits of cooperation.

I enjoyed reading the paper. In fact, the paper is rich with theoretical implications, and the mechanisms of collapse and rescue of cooperation presented here are novel. I have several suggestions to improve the paper, as described below.

[1] Reference to previous works on evolution of cooperation in a dynamic-network setting is unfortunately not rich enough (only citations 30-32). In particular, many studies have intensively investigated the effect of dynamic linking (or dynamic link-weight adjustment) on evolution of cooperation. Those works typically assume that the link is maintained (or the link weight is increased) when one benefits from the interaction with the partner, and otherwise the link is broken (or the link weight is decreased). To list a few, Huang, Zheng & Yang (2015; Scientific Reports), Fu, Hauert, Nowak, & Wang (2008;PRE), and Skyrms and Pemantle (2000; PNAS). Consider citing those (and other) papers.

I thank the reviewer for these comments and references. As I discussed in the original submission, most previous models with dynamic networks (including the ones suggested above by the reviewer) deal with some sort of partner choice (where links are reinforced according to whether the interaction was rewarding or not), and as such go in a different direction from the current paper. Specifically, the papers the reviewers mention all look at cases of the links between a pre-existing (and fixed) set of individuals changing, which is a different situation than the dynamic networks I model here. Nonetheless, I agree this particular type of models is worth mentioning, and I have added a few more references as suggested by the reviewer in the introduction.

[2] The author finds that the effect of p_n is quite marginal (page 7). However, I naively expect that inheriting links from one's parent, especially when cooperation is prevalent in the population, should be very beneficial, because it is highly likely that his/her parent would have many cooperative neighbors. Please provide more explanations to that.

The reviewer is correct that this finding merits a bit more comment, and I now comment on it both in the Results and Discussion sections. Briefly, the reason is that our intuition stems from models of social evolution where everyone has the same number of connections (e.g., in patch-structured models). In that context, inheriting more connections would mean making fewer random connections, and therefore increase the mean assortment between your partners. But in my model, there is no such trade-off, and in fact for fixed p_r , higher p_n slightly reduces the mean assortment between individuals (e.g., see new SI Figure 1 in the current version). In the limit of very high p_n , the network is so densely connected that there is very little assortment between cooperators, which is why cooperation collapses at that limit.

[3] Model (page 4): Because the author's model considers probabili-

ties of link-inheritance and random-connection PER INDIVIDUAL, the absolute number of connections increases with increased population size, N . This makes me wonder whether the author’s result is scale-free or not, because many previous studies have shown the importance of absolute neighborhood size (see, for example, reference 27). Put differently, I wonder if the result is qualitatively unchanged if N becomes two/five/ten times larger, or so. I naively expect that this would increase the neighborhood size and would considerably disfavor cooperation. Is that right?

This is a fair point, also raised by Reviewer 2: I now include results that replicate the previous analyses for $N=200$ and $N=500$ in the supplement, which show that network size does not affect the qualitative patterns of mean cooperation and linking traits. The main change in mean values with population size is (as the reviewer correctly notes) that the linking probabilities are per-capita, and therefore with increasing network size the relevant range (especially the threshold p_r required to sustain cooperation) changes. Further, I now include more results in the supplementary material with larger networks, where persistent polymorphisms or cycling can happen.

[4] The rate of strategy evolution is controlled by delta, whereas the rate of linking probability evolution is controlled by μ_1 and σ ’s in this paper. I wonder if changing their relative balance could change the results. In particular, can we observe a cyclic behavior of p_r increasing, cooperation collapsing, followed by the decrease of p_r , and by re-emergence of cooperation? Or is evolution always in one way, in the sense that, once the increased level of p_r undermines cooperation, cooperation never recovers evolutionarily?

First, a clarification: delta in the model is the strength of selection that applies to selection on both strategy and linking traits, as both traits affect the payoff, and delta determines how payoff translates into fitness. That said, the reviewer’s intuition is correct in that we can observe cycling when there is a cost of linking, where cooperation can select for higher p_r , which leads to collapse, which (due to the costs of making connections) leads to lower p_r . In smaller networks this cycling is masked by the stochasticity of the simulation and long-term averages are meaningful (as measuring how long the population spends at high and low cooperation states), but it can be more readily observed in bigger networks. I now include these results in the supplementary material.

[5] “ 1_i ” in eqs.(1) and (4) should be “ p_i ”.

Thank you. Fixed.

[6] “work in exactly the same way” (lines 467-468 in SI) is ambiguous. I think the author wants to point out that both the C -term and the C_{link} -term are proportional to the number of connections, $d_i(t)$ in eq.(SI-1) whereas it is not the case in eq.(4) in the main text. Please add more words here.

The reviewer is right that this requires more explanation. I added a more explicit discussion of why cooperation always collapses in the PD game in the Discussion and the SI.

[7] Clarify parameters used in each subsection: in the “Fixed linking probabilities” section I guess $C_{\text{link}}=0$. In the “Coevolution of linking probabilities ...” subsection I guess $C_{\text{link}}=D=0$.

I clarified these parameters.

[8] I occasionally find minor grammatical errors. For example, “the higher linking costs have to be [to] maintain it” (abstract), “..., mostly independent[ly] of the probability” (page 4, top). Please review the whole manuscript again and clean them off.

I thank the reviewer for the careful reading. I read the manuscript over and corrected these and any other grammatical errors I could find.

REVIEWERS' COMMENTS:

Reviewer #1 (Remarks to the Author):

The author has done a very good job with the revision. I especially like the addition of the final Results section, "Exogenously high social inheritance can rescue cooperation." I think this adds a lot to the paper. My recommendation is acceptance in its current form. I do want to note that I found a few typos, listed below.

Line 116: Missing bracket. "when social inheritance is high 36]"

Lines 229-231: "In other words, even though cooperation can be rescued by costs of social connections, the victory may prove pyrrhic." Typo, should be "pyrrhic".

Lines 243-245: "In general, synergistic payoffs, together with some costs of linking promote cooperation and increase mean fitness 5(c), but result in very"

Sentence fragment. You appear to have cut off some of this sentence. I notice that it's a new one not in the original submission. Curious to know what the end is.

Lines 300-301: "This logic behind this phenomenon..." First "This" should be "The"

Line 401: "Schank" is misspelled as "Shank"

Reviewer #2 (Remarks to the Author):

The author considerably improved the presentation of the work. The model and results sections are much clearer than those of the previous version. More analysis is presented in the Supplementary Information (SI) and it results to be very useful to better understand the dynamics of the systems. The network size analysis is robust, showing that presented results also hold for larger system sizes. The discussion on the degree influence on cooperation in the SI increases the value of the contribution, although it can be better investigated in future works. The author satisfactorily addresses previous concerns on the convergence of the results. The discussion is really well argued and well structured. Overall, the study can be considered for publication after addressing the comments below.

Regarding my previous comment on the number of model parameters, I underline the fact that the study does not include all possible sensible parameter combinations (of course, avoiding all the unavoidable ones). Having linking probabilities, synergy, and linking costs, as principal parameters (obtaining $2 \times 2 \times 2 = 8$ possible scenarios to be studied), the author does not study the behavior of the system when probabilities are fixed and links are costly (it is slightly considered in Fig. 6 when no synergy is present and only p_r evolves). Also, the system with evolving probabilities, synergy and no cost is only present in one line of simulations of Fig. 5. These complementary results can be briefly addressed in the text without further simulations but just giving to the reader an idea of the possible outcomes.

Another important comment is related to the Prisoner's Dilemma (PD) results. It appears that the mechanisms responsible for the collapse and the recover of cooperation only work for the n-player game, i.e. the coauthor game, and not for the PD game, or at least, not for the version the author proposes in the SI. The PD results are presented only for the strong selection limit and, apparently, without the synergistic parameter D . The game values are also pretty different from the ones of the main text for the coauthor game. This does not allow a clear comparison between the two games (Fig. SI 11 and Fig. 1). I would suggest to only focus on the coauthor game, since the PD results can be very misleading and can be removed.

It would be better to use the same color scale for Fig. SI 3c and SI 4c, perhaps the one of Fig. SI 3c for a better comparison.

The abstract is almost the same of the previous version. The last three sentences should be rephrased: "My model shows (that) cooperation...", "...have to be maintain it...". It is not specified which coevolutionary dynamics should constrain cooperation.

The two probabilities p_n and p_r appear to be directly linked to the average degree of the population ($\langle k \rangle$) and its size (N). Can the author discuss their actual values compared to the system size/average degree? Also, how are they actually implemented ("it connects to other individuals that are not connected to its parent with probability p_r " can mean that with that probability it connects with all of them, which I assume it is not the case)? When p_r increases, is it reasonable to expect that also the average degree increases?

The Simulations section before the Results is too technical and it may be moved to the SI. Especially the part of the coding. The parameter values should be instead included in each figure caption. It is also slightly misleading to say that the population evolves for 20 generations without selection in order to obtain the required network structure of [36]. The 20 steps can be simply included into the network construction process, as already explained in [36] and just referring to it.

Reviewer #3 (Remarks to the Author):

The author has responded to all my comments appropriately. There are no remaining issues, except for the typos below. I congratulate the author on this very nice paper!

L170: where-> were

L227: Italicize B .

L231: phyrlic -> pyrrhic

L245: the sentence does not finish correctly.

L263: "equals to" or "is equal to"

L338: favoring

Response to Reviewers

For: “Collapse and rescue of cooperation in evolving dynamic networks” Erol Akçay, final version submitted to Nature Communications

Erol Akçay

I thank all three reviewers and the editor for detailed attention to this manuscript, which helped me improve it greatly. Below, I detail the changes made in response to the final round comments of the reviewers:

Associate Editor:

I thank the Associate Editor for their careful reading of the manuscript and her suggestions and corrections, which I have followed. One issue that bears remarking here is the inclusion of the Prisoner’s Dilemma (PD) results in the Supplementary Information. As I also discuss in my response to Reviewer 2, I am including the PD results mainly as a useful comparison with the (to me, more realistic) model of the co-author game. This comparison is interesting since most social evolution models on networks use the PD payoff model, and the fact that most results with the coauthor game carry over to the PD model is a useful robustness check. The one result that differs sharply between the payoff functions (that costs of connections does not rescue cooperation) is also of interest but not as central to the main point of the paper. Therefore, I still include the PD game in the Supplementary Information.

Reviewer 1:

The author has done a very good job with the revision. I especially like the addition of the final Results section, “Exogenously high social inheritance can rescue cooperation.” I think this adds a lot to the paper. My recommendation is acceptance in its current form. I do want to note that I found a few typos, listed below.

I thank the reviewer for their kind comments, and also for their careful reading. I fixed all the typos below, and attempted to find others.

Line 116: Missing bracket. “when social inheritance is high 36]”

Lines 229-231: “In other words, even though cooperation can be rescued by costs of social connections, the victory may prove pyrrhic.” Typo, should be “pyrrhic”.

Lines 243-245: “In general, synergistic payoffs, together with some costs of linking promote cooperation and increase mean fitness 5(c), but result in very” Sentence fragment. You appear to have cut off some of this sentence. I notice that it’s a new one not in the original submission. Curious to know what the end is.

Apologies for the unintended suspense: it was “. . . sparsely connected networks.”

Lines 300-301: “This logic behind this phenomenon. . .” First “This” should be “The”

Line 401: “Schank” is misspelled as “Shank”

Reviewer 2:

The author considerably improved the presentation of the work. The model and results sections are much clearer than those of the previous version. More analysis is presented in the Supplementary Information (SI) and it results to be very useful to better understand the dynamics of the systems. The network size analysis is robust, showing that presented results also hold for larger system sizes. The discussion on the degree influence on cooperation in the SI increases the value of the

contribution, although it can be better investigated in future works. The author satisfactorily addresses previous concerns on the convergence of the results. The discussion is really well argued and well structured. Overall, the study can be considered for publication after addressing the comments below.

I thank the reviewer for their comments and suggestions.

Regarding my previous comment on the number of model parameters, I underline the fact that the study does not include all possible sensible parameter combinations (of course, avoiding all the unavoidable ones). Having linking probabilities, synergy, and linking costs, as principal parameters (obtaining $2 \times 2 \times 2 = 8$ possible scenarios to be studied), the author does not study the behavior of the system when probabilities are fixed and links are costly (it is slightly considered in Fig. 6 when no synergy is present and only p_r evolves). Also, the system with evolving probabilities, synergy and no cost is only present in one line of simulations of Fig. 5. These complementary results can be briefly addressed in the text without further simulations but just giving to the reader an idea of the possible outcomes.

The reviewer is correct that I do not explicitly consider all potential scenarios, but the only major exception is fixed linking probabilities and linking costs. The reason that I did not separately consider this scenario is that there is no reason to expect these results will be any different than the setting without costs, at least under weak selection. As I discuss in the main text, even without variation in linking probabilities, network structure can be affected by selection, including selection due to costs of linking. But such selection will only be possibly relevant under strong selection: well-known results in social evolution theory (and my own simulations) show that under weak selection population structure is well-approximated by the neutral selection. Therefore, under weak selection, the effects of costs of linking will be negligible. This can be seen in Figure 1 in this document, which corresponds to the same scenario as the middle panel of Figure 1 of the manuscript but with costs of connection ($C_l = 0.2$). Under strong selection, it is true that costs of linking can have an effect on the network structure even with fixed linking probabilities, as in Cavaliere et al 2012 (whose results I was only able to replicate with very high cost and benefit values, effectively corresponding to strong selection). The effect would be to reduce the mean connectivity of the network relative to neutral structure, which all things being equal would favor cooperation for a wider range of p_n and p_r values, but would not change the qualitative patterns. I now make these points in the discussion.

Figure 1: Right hand panel is the middle Panel of Figure 1 of the main text, left-hand panel the same conditions but with costs of connection.

Another important comment is related to the Prisoner's Dilemma (PD) results. It appears that the mechanisms responsible for the collapse and the recover of cooperation only work for the

n-player game, i.e. the coauthor game, and not for the PD game, or at least, not for the version the author proposes in the SI. The PD results are presented only for the strong selection limit and, apparently, without the synergistic parameter D . The game values are also pretty different from the ones of the main text for the coauthor game. This does not allow a clear comparison between the two games (Fig. SI 11 and Fig. 1). I would suggest to only focus on the coauthor game, since the PD results can be very misleading and can be removed.

Having considered the reviewer's suggestion, I decided *not* drop the PD game. I believe that the PD game does make a useful comparison to the coauthor game, and the contrast in the results are informative (see also response to the Associate Editor above). In particular, the mechanism for collapse of cooperation is general and will apply in any game where an additional link to a cooperative individual will provide positive benefits, including the Prisoner's Dilemma game. The reviewer is correct that the mechanism for rescue I investigate, fixed costs of linking, does not work for the PD game but that has nothing to do with weak vs. strong selection. In this version, I replace the strong selection simulation results for the PD game with the weak selection versions (which I agree are more comparable to the main text): they show identical patterns to strong selection. The theoretical argument (included in the discussion) applies just as well to weak selection as it does to strong selection.

As for the comparison of the payoff parameters, it is important to note that the benefit in the coauthor game is divided by the degree of an individual, while the cost in the PD game is multiplied by the degree (in fact, this is the only difference between the game structures; neither can be said to be more of an N-person game than the other). Hence, to get the correct order-of-magnitude match between the benefit from a cooperator to a partner and the cost to the cooperator in the two games, B and C in the PD should be of order $1/d$ times the B and C in the coauthor game. This is why I use lower values for B and C in the PD games (roughly, $1/5$ th) as in the coauthor game. I also note that, none of my results are sensitive to parameter values (an are presented for a range of parameter values). Therefore, the qualitatively different results between the coauthor and PD games are not driven by a mismatch of parameters, and are informative, so I kept the PD results in the SI.

It would be better to use the same color scale for Fig. SI 3c and SI 4c, perhaps the one of Fig. SI 3c for a better comparison.

I followed the reviewer's suggestion, using the color range of SI FIG 4C for SI Fig 3C, which has a much smaller range than 4C.

The abstract is almost the same of the previous version. The last three sentences should be rephrased: "My model shows (that) cooperation...", "...have to be maintain it...". It is not specified which coevolutionary dynamics should constrain cooperation.

I edited the abstract to reflect my main points more clearly.

The two probabilities p_n and p_r appear to be directly linked to the average degree of the population (p_n) and its size (p_r). Can the author discuss their actual values compared to the system size/average degree? Also, how are they actually implemented ("it connects to other individuals that are not connected to its parent with probability p_r " can mean that with that probability it connects with all of them, which I assume it is not the case)? When p_r increases, is it reasonable to expect that also the average degree increases?

I clarified this point in the model description: p_n and p_r are *per individual* probabilities to connecting to each individual that is connected and unconnected to the parent, respectively. The relationship between the linking probabilities and the expected degree and clustering coefficient under neutrality is worked out analytically in Ilany and Akcay, 2016.

The Simulations section before the Results is too technical and it may be moved to the SI. Especially the part of the coding. The parameter values should be instead included in each figure caption. It is also slightly misleading to say that the population evolves for 20 generations without selection in order to obtain the required network structure of [36]. The 20 steps can be simply included into the network construction process, as already explained in [36] and just referring to

it.

I do not follow why saying that I ran the network for the 20 generations under neutrality is misleading, since that is in fact what I do. The point of this burn in period is just to ensure that we are starting with a network representative of the model under neutrality, to guard against any transient effects that might arise from starting with a random network. In the previous paper, we showed that 20 generations is enough to ensure convergence to the stationary distribution of the stochastic network dynamics. I should emphasize again that the network keeps changing throughout the dynamics with each birth and death event. With regard to the suggestion to move the simulation details to the SI, since the Methods section is already typeset in smaller print at the end of the paper, I felt that the reviewer's (presumed) concern about the technical description breaking the flow of the paper is mitigated. Therefore, I decided to keep this short description here.

Reviewer 3:

The author has responded to all my comments appropriately. There are no remaining issues, except for the typos below. I congratulate the author on this very nice paper!

Thank you for the kind comments and for finding the typos, which are fixed.

L170: where-> were

L227: Italicize B.

L231: phyrrie -> pyrrhic

L245: the sentence does not finish correctly.

L263: "equals to" or "is equal to"

L338: favoring